# Epidemiology of type 2 diabetes remission in Scotland in 2019: A cross-sectional population-based study

**Mireille Captieux**[1]*, **Kelly Fleetwood**[1], **Brian Kennon**[2], **Naveed Sattar**[3], **Robert Lindsay**[3], **Bruce Guthrie**[1], **Sarah H. Wild**[1], on behalf of the Scottish Diabetes Research Network Epidemiology Group

**1** Usher Institute, The University of Edinburgh, Edinburgh, United Kingdom, **2** Queen Elizabeth University Hospital, Glasgow, United Kingdom, **3** Institute of Cardiovascular and Medical Sciences, The University of Glasgow, Glasgow, United Kingdom

* mireille.captieux@ed.ac.uk

## Abstract

### Background

Clinical pathways are changing to incorporate support and appropriate follow-up for people to achieve remission of type 2 diabetes, but there is limited understanding of the prevalence of remission in current practice or patient characteristics associated with remission.

### Methods and findings

We carried out a cross-sectional study estimating the prevalence of remission of type 2 diabetes in all adults in Scotland aged ≥30 years diagnosed with type 2 diabetes and alive on December 31, 2019. Remission of type 2 diabetes was assessed between January 1, 2019 and December 31, 2019. We defined remission as all HbA1c values <48 mmol/mol in the absence of glucose-lowering therapy (GLT) for a continuous duration of ≥365 days before the date of the last recorded HbA1c in 2019. Multivariable logistic regression in complete and multiply imputed datasets was used to examine characteristics associated with remission. Our cohort consisted of 162,316 individuals, all of whom had at least 1 HbA1c ≥48 mmol/mol (6.5%) at or after diagnosis of diabetes and at least 1 HbA1c recorded in 2019 (78.5% of the eligible population). Over half (56%) of our cohort was aged 65 years or over in 2019, and 64% had had type 2 diabetes for at least 6 years. Our cohort was predominantly of white ethnicity (74%), and ethnicity data were missing for 19% of the cohort. Median body mass index (BMI) at diagnosis was 32.3 kg/m². A total of 7,710 people (4.8% [95% confidence interval [CI] 4.7 to 4.9]) were in remission of type 2 diabetes. Factors associated with remission were older age (odds ratio [OR] 1.48 [95% CI 1.34 to 1.62] P < 0.001) for people aged ≥75 years compared to 45 to 54 year group), HbA1c <48 mmol/mol at diagnosis (OR 1.31 [95% CI 1.24 to 1.39] P < 0.001) compared to 48 to 52 mmol/mol), no previous history of GLT (OR 14.6 [95% CI 13.7 to 15.5] P < 0.001), weight loss from diagnosis to 2019 (OR 4.45 [95% CI 3.89 to 5.10] P < 0.001) for ≥15 kg of weight loss compared to 0 to 4.9 kg weight gain), and previous bariatric surgery (OR 11.9 [95% CI 9.41 to 15.1] P < 0.001).

**Data Availability Statement:** The authors are not permitted to share the data that support the findings of this study directly. Data from the Scottish Care Information – diabetes database are

available to accredited researchers who receive approval for data access through a data safe haven from the NHS Scotland Public Benefit and Privacy Panel for Health and Social Care: https://www.informationgovernance.scot.nhs.uk/pbpphsc/.

**Funding:** MC was was funded by Chief Scientist Office CAF 18/12 (https://www.cso.scot.nhs.uk/personal-awards-initiative/clinical-academic-fellowships/). The funders had no role in the study design, data collection, analysis, decision to publish, or preparation of the manuscript.

**Competing interests:** I have read the journal's policy and the authors of this manuscript have the following competing interests: BK is national lead for diabetes and chair of the Scottish Diabetes Group which sits directly within the Clinical Priorities team at Scottish Government, member of the Type 2 Diabetes Prevention Oversight group which is a Scottish Government related group, speciality adviser to Chief Medical Officer for diabetes and endocrine. RL has served on advisory boards with Novo Nordisk, Lily and Servier only NS has consulted for Amgen, AstraZeneca, Boehringer Ingelheim, Eli Lilly, Merck Sharp & Dohme, Novartis, Novo Nordisk, Pfizer, and Sanofi, and received grant support from Boehringer Ingelheim, Novartis and Roche, outside the submitted work.

**Abbreviations:** 2-hr PG, two-hour plasma glucose; ADA, American Diabetes Association; AHEAD, Action for Health in Diabetes; AIC, Akaike information criterion; BMI, body mass index; CI, confidence interval; COVID-19, Coronavirus Disease 2019; DiRECT, Diabetes Remission Clinical Trial; DPP-4, dipeptidyl peptidase 4; DUK, Diabetes UK; FPG, fasting plasma glucose; GLP-1, glucagon-like peptide-1 receptor; GLT, glucose-lowering therapy; ICD-10, International Classification of Diseases-10th revision; IDF, International Diabetes Federation; MAR, missing at random; MCAR, missing completely at random; MICE, Multiple Imputation by Chained Equations; NHS, National Health Service; NMAR, not missing at random; OPCS-4, Office of Population Censuses and Surveys Classification of Interventions and Procedures version 4; OR, odds ratio; PCDS, Primary Care Diabetes Society; SCI-Diabetes, Scottish Care Information-Diabetes; SGLT-2, sodium glucose cotransporter-2; SIGN, Scottish Intercollegiate Guidelines Network; SIMD, Scottish Index of Multiple Deprivation; STROBE, Strengthening The Reporting of OBservational Studies in Epidemiology; WHO, World Health Organization.

Limitations of the study include the use of a limited subset of possible definitions of remission of type 2 diabetes, missing data, and inability to identify self-funded bariatric surgery.

## Conclusions

In this study, we found that 4.8% of people with type 2 diabetes who had at least 1 HbA1c ≥48 mmol/mol (6.5%) after diagnosis of diabetes and had at least 1 HbA1c recorded in 2019 had evidence of type 2 diabetes remission. Guidelines are required for management and follow-up of this group and may differ depending on whether weight loss and remission of diabetes were intentional or unintentional. Our findings can be used to evaluate the impact of future initiatives on the prevalence of type 2 diabetes remission.

## Author summary

### Why was this study done?

- Feasibility of diabetes remission has been demonstrated in research settings and after bariatric surgery, but we do not know how many people in the general population achieve remission of type 2 diabetes.

- Informed decisions need to be made about which people are most likely to achieve and maintain remission; to do this, we need to better understand the characteristics of people who are currently in remission.

- Estimating the prevalence of remission of type 2 diabetes in Scotland in 2019 creates a baseline to evaluate the impact of future initiatives to support remission and for future studies of duration of remission and effect on risk of complications of diabetes.

### What did the researchers do and find?

- We calculated how many people were in remission of type 2 diabetes in 2019 in Scotland from a national type 2 diabetes register. This register contains 99% of people with diabetes in Scotland.

- We described the characteristics of people who were in remission of type 2 diabetes compared to people who were not in remission and created a mathematical model that shows the probability of achieving remission in 2019 based on these characteristics.

- We found that about 1 in 20 of people with type 2 diabetes in the study population were in remission of type 2 diabetes.

- Compared to people who did not achieve remission, people in remission of type 2 diabetes tended to be older; have a lower HbA1c at diagnosis; have never taken any glucose-lowering medication; have lost weight since the diagnosis of diabetes; and have had bariatric surgery.

**What do these findings mean?**

- There is a sizeable proportion of people who achieve remission of type 2 diabetes outside research trials and without bariatric surgery. These people should be recognised and coded appropriately so they can be supported by their clinicians. The clinical progress of these people can now be followed by researchers.

- People who have not yet been prescribed drugs to treat diabetes may be the most appropriate group for clinicians to initiate discussions around remission and weight management options.

- Guidelines for supporting people who achieve remission of diabetes must recognise differences between people that lose weight intentionally and those that lose weight because of severe illness. Clinicians also need greater clarity on how to manage older or frailer people who achieve remission criteria.

## Introduction

There were an estimated 463 million people with diabetes in the world in 2019, of whom 90% to 95% have type 2 diabetes [1]. By 2045, it is estimated that there will be 700 million people in the world with diabetes. Drivers for the global rise in diabetes prevalence include increasing numbers of people aged >65 years of age; urbanisation; increasing prevalence of obesity; and improved survival of people with diabetes [2,3]. Remission of type 2 diabetes (defined broadly as the achievement of normal glycaemic measures without glucose-lowering therapy (GLT)) may be one way to flatten this upward trend. Position statements or recommendations for practice have been published by the Primary Care Diabetes Society (PCDS) [4], the International Diabetes Federation (IDF) [5], and a multidisciplinary group of experts [6] in 2019, 2017, and 2009, respectively. The PCDS states that "remission can be achieved when a person with type 2 diabetes achieves 1. Weight loss; 2. HbA1c <48 mmol/mol (6.5%) or FPG <7.0 mmol/l (126mg/dL) on two occasions separated by six months; 3. Following complete cessation of all GLT." [4] (p. 74). The IDF states that "remission is defined by most guidelines as an HbA1c below 6% (42 mmol/mol) without medication for 6 months or more" [5] (p. 21)." Buse and colleagues define remission as "achieving glycaemia below the diabetic range in the absence of active pharmacologic or surgical therapy." Three types of remission are explicitly defined: partial remission, complete remission, and prolonged remission (cure) with a minimum duration of 1 year for partial and complete remission [6] (p. 2134). Riddle and colleagues define remission as "HbA1c <6.5% (48 mmol/mol) measured at least 3 months after cessation of glucose-lowering pharmacotherapy" (p. 1) [7] (at least 6 months after starting a lifestyle intervention) [7]. We have previously shown that there were at least 96 unique definitions of diabetes remission used in the research literature from 2009 to 2020 [8].

In the early 1990s, remission was demonstrated in people with type 2 diabetes after bariatric surgery [9]. This challenged the perception of type 2 diabetes as a chronic progressive disease. Two recent trials have additionally shown that it is also possible to achieve remission of type 2 diabetes through weight loss using very low calorie diets [10,11]. The Diabetes Remission Clinical Trial (DiRECT) was the first trial to use a low calorie diet intervention to assess type 2 diabetes remission as a primary outcome. Participants were between 20 and 65 years of age, with body mass index (BMI) 27 to 45 kg/m$^2$, and within 6 years since diabetes diagnosis. After 2

years of follow-up, they reported remission of 36% in their intervention group and 3% in their control group [12]. This nonsurgical approach has the potential to make remission of type 2 diabetes more widely feasible without the adverse long-term effects of bariatric surgery. Since the publication of these trials, achieving remission of type 2 diabetes has been identified as a top priority by people with diabetes and their carers [13]. United Kingdom governments have recently included remission of type 2 diabetes in their long-term type 2 diabetes frameworks [14,15], and the American Diabetes Association (ADA) Standards of Medical Care has, for the first time, included guidance on prescribing very low calorie diets to improve glycaemic control and promote remission of diabetes [16]. The National Health Service (NHS) in England and Scotland are currently introducing the use of very low calorie diets for obese people with type 2 diabetes in routine clinical care [14]. The recent finding that type 2 diabetes is independently associated with increased odds of death with Coronavirus Disease 2019 (COVID-19) may further accelerate interest in achieving remission of type 2 diabetes [17].

Although remission of type 2 diabetes has been observed in trial settings and following bariatric surgery, it is unclear how common remission is in normal care. Estimating prevalence of type 2 diabetes remission is needed to inform allocation of resources and creation of new clinical pathways to support this group of people to stay in remission. Prevalence estimates also provide context for clinical decision-making, for example, identifying groups for whom remission is most likely to be achievable in order to target limited resources for intensive lifestyle management. Additionally, evaluating the impact of new clinical pathways to support remission of type 2 diabetes requires understanding of patterns of remission prior to introduction of new services.

Our aims were to estimate the prevalence of remission of type 2 diabetes in Scotland in 2019 and to compare the characteristics of people with and without type 2 diabetes remission at the population level.

## Methods

### Study design and data sources

We used a cross-sectional whole population study design to estimate the prevalence of remission of type 2 diabetes in Scotland in 2019 and describe the characteristics of people in remission and people not in remission. The population was identified from The Scottish Care Information-Diabetes (SCI-Diabetes) registry, which is a population-level diabetes registry derived from NHS primary and secondary care data. SCI-Diabetes contains demographic and clinical data for over 99.5% of people with a diagnosis of diabetes in Scotland [18]. Type of diabetes in SCI-Diabetes is validated for research purposes using an algorithm combining clinical coding, treatment, and age of onset [2,19]. This was an exploratory cross-sectional analysis for which there was no prespecified protocol. We reported our findings using the Strengthening The Reporting of OBservational Studies in Epidemiology (STROBE) Checklist for cross-sectional studies (S1 Table).

We extracted data for people with a diagnosis of type 2 diabetes who were registered as alive and resident in any Scottish health board on December 31, 2019. From this population, we selected people who were aged ≥30 years who were diagnosed with type 2 diabetes between January 1, 2004 (when registration to SCI-Diabetes was near complete across Scotland) and December 31, 2018 [2]. We excluded the small number of people aged <30 years at diagnosis of type 2 diabetes to mitigate against the inadvertent inclusion of people with type 1 diabetes. As a key criterion for remission was HbA1c <48 mmol/ mol (6.5%), the main analysis excluded people without HbA1c readings in 2019 or people with no record of HbA1c readings ≥48 mmol/mol (6.5%) at any point from diagnosis to December 31, 2019. In our systematic

review, we discussed the inconsistency of current diabetes diagnostic criteria with existing remission criteria [8]. In order to manage this issue, we included only people who recorded an HbA1c in the diabetes range (≥48 mmol/mol) at some point from diagnosis to 2019, accepting that a proportion of these people may have been initially diagnosed on the basis of fasting plasma glucose (FPG) or two-hour plasma glucose (2-hr PG) alone (although they all subsequently went on to record HbA1c readings ≥48 mmol/mol). People with no record of an HbA1c ≥48 mmol/mol (6.5%) after diagnosis of diabetes or no HbA1c recorded during 2019 were excluded initially but included in sensitivity analyses.

## Outcome

Remission of type 2 diabetes was assessed between January 1, 2019 and December 31, 2019. Although there are national and organisational consensus definitions, as yet there is no international consensus for defining remission, resulting in considerable heterogeneity in remission definitions used in research [8]. For the purposes of this analysis, we defined remission on the basis of all HbA1c values being <48 mmol/mol in the absence of GLT for a duration of ≥365 days before the date of the last recorded HbA1c in 2019, that is on the basis of at least 2 HbA1c values <48 mmol/mol at an interval of at least 365 days.

## Covariates

Covariates were chosen based on previous literature [11,20] and clinical relevance. Demographic variables recorded at date of diagnosis of type 2 diabetes were sex and age (categorised into the following groups: 30 to 44, 45 to 54, 55 to 64, 65 to 74, and 75 and over years). The closest record ±90 days around the date of diagnosis was used to define smoking status. Socioeconomic status was defined using the 2016 Scottish Index of Multiple Deprivation (SIMD). This is an area-based deprivation measure that is the national standard for assessing socioeconomic status in research in Scotland [21] (categorised into SIMD quintiles where group 1 is the most deprived and group 5 the least deprived fifth of the population). Continuous covariates at diagnosis were defined as the median of all values recorded within a ± 90-day window around diagnosis (HbA1c at diagnosis; BMI at diagnosis; and weight at diagnosis). Median HbA1c at diagnosis was additionally categorised into 5 groups based on clinical targets outlined by the Scottish Intercollegiate Guidelines Network (SIGN) [22]: <48 mmol/mol (<6.5%); 48 to 52 mmol/mol (6.5% to 6.9%); 53 to 63 mmol/mol (7% to 7.9%); 64 to 85 mmol/mol (8.0% to 9.9%); and ≥86 mmol/mol (≥10%). BMI at diagnosis was categorised into 5 groups based on a simplified World Health Organization (WHO) classification: <18.5 kg/m$^2$; 18.5 to 24 kg/m$^2$; 25 to 29 kg/m$^2$, 30 to 34 kg/m$^2$, 35 to 39 kg/m$^2$; and ≥40 kg/m$^2$.

Duration of type 2 diabetes on December 31, 2019 was calculated using date of diagnosis and categorised into 2 groups: ≤6 years and >6 years. Age in 2019 was calculated by subtracting date of birth from December 31, 2019, and continuous covariates in 2019 were defined as the median of values recorded in the 365 days before December 31, 2019. For age and continuous covariates, the same categories as baseline were applied. Weight change was defined as the value recorded in 2019 subtracted from the value recorded closest to diagnosis of diabetes. We defined GLT as any combination of metformin, sulphonylureas, thiazolidinediones, dipeptidyl peptidase 4 (DPP-4) inhibitors, sodium glucose cotransporter-2 (SGLT-2) inhibitors, glucagon-like peptide-1 receptor (GLP-1) agonists, and insulin. Prescription data were defined using Read Codes, the clinical terminology system widely used in UK general practice. We identified previous bariatric surgery from NHS Scotland hospital admission data by identifying people with codes for relevant gastrointestinal surgery (based on the combination of Office of Population Censuses and Surveys Classification of Interventions and Procedures version 4

(OPCS-4) codes and the International Classification of Diseases-10th revision (ICD-10) diagnostic code for obesity used by Public Health Scotland) (see S2 Table for diagnosis and procedure codes). We identified diagnostic codes for severe comorbidity that could cause weight loss including previous diagnosis of dementia, end-stage kidney disease, liver cirrhosis, metastatic cancer, and cancer (excluding nonmelanoma skin cancer) with a 5-year look-back period applied to both metastatic cancer and cancer diagnoses (see S3 Table).

## Statistical analysis

The prevalence of type 2 diabetes remission in 2019 was assessed for people who had at least 1 HbA1c ≥48 mmol/mol (6.5%) at or after diagnosis of diabetes and had at least 1 HbA1c recorded in 2019. People with no record of an HbA1c ≥48 mmol/mol (6.5%) after diagnosis of diabetes or no HbA1c recorded during 2019 were excluded initially but included for sensitivity analyses for estimates of the prevalence of remission. The first sensitivity analysis assumed that none of the people without an HbA1c reading in 2019 were in remission. The second sensitivity analysis assumed all people who had never had an HbA1c ≥48 mmol/mol were in remission.

Proportions of missing data in variables of interest were examined, and each covariable in our dataset was analysed using a univariable logistic regression model to estimate the odds of missing data within that variable and therefore explore likely mechanism of missingness. Odds ratios (ORs) over 1 indicate higher odds of missingness relative to the reference category (see S4 Table for details) [23]. This analysis of distribution of missing data showed that missingness was associated with some of the variables. For example, people with dementia had high odds of missing data compared to people who did not have dementia (OR 3.02 [95% confidence interval [CI] 2.70 to 3.37] $P < 0.001$) (S4 Table). Therefore, data were not missing completely at random (MCAR) and were either MAR or not missing at random (NMAR). We managed missing data using multiple imputation using the Multiple Imputation by Chained Equations (MICE) method based on the assumption of the data being missing at random (MAR) [24]. A total of 30 imputed datasets were used based on the greatest percentage of incomplete cases in the dataset [25].

Associations between demographic and clinical covariates and type 2 diabetes remission were examined using univariable and multivariable logistic regression analysis on both complete case and multiple imputation datasets. Age and sex were included a priori in the models. Two-way interactions between each of age, sex, HbA1c at diagnosis, GLT prescription, weight change, and bariatric surgery were investigated; however, none improved overall model fit and therefore were not included in the final model. The statistical significance of each variable was tested at the 99% level by comparing the model fit with the variable of interest to the same model without that variable [24]. Model fit was assessed using the Akaike information criterion (AIC) [26]. A sensitivity analysis was carried out to explore the effect of excluding people with severe comorbidity (as defined above) that could cause unintentional weight loss.

Statistical significance was defined as $P < 0.05$ with 95% CIs reported. Data were analysed using R version 3.4.4.

Approval for creation of the linked dataset used in this analysis was obtained from the Scotland A multicentre research ethics committee, reference number 11/AL/0225.

## Patient and public involvement

Patients and the public were not involved in the design, conduct, or reporting of this research. However, the Diabetes UK (DUK)–James Lind Alliance Priority Setting Partnership has recently identified remission of type 2 diabetes as the top shared priority among people living

with diabetes and their carers and healthcare professionals [13], and the findings of the work are being shared with patient groups, Diabetes Scotland, and DUK.

## Results

There were 206,856 people aged ≥30 years with type 2 diabetes diagnosed in Scotland between January 1, 2004 and December 31, 2018 who were alive and resident in any Scottish health board on December 31, 2019. Of these, 35,321 (17%) did not have any HbA1c records for 2019 and a further 9,219 (4.5%) never had a record of an HbA1c value ≥48 mmol/mol (6.5%), leaving 162,316 people for the primary analysis, of whom 117,048 (72%) had complete data. Fig 1 provides a description of the cohort selection for different aspects of the analysis.

Of the 162,316 patients eligible for the primary analysis, 7,710 (4.8% [95% CI 4.7 to 4.9]) were in remission in 2019. Three sensitivity analyses for estimates of the prevalence of remission were carried out. First, assuming that none of the 35,321 patients without an HbA1c reading in 2019 was in remission and adding them to the denominator resulted in an estimated prevalence in the whole cohort of 206,856 of 3.7% (95% CI 3.6 to 3.8). Second, inclusion of the 9,219 patients who had never had an HbA1c ≥48 mmol/mol in both numerator and denominators meant that estimated prevalence of diabetes remission in the 171,535 patients with at least 1 HbA1c in 2019 was 9.9% (95% CI 9.7 to 10.0). Third, changing the remission duration cut point from 12 to 10 months to also include people whose last HbA1c was between 10 and 12 months changed the prevalence estimate to 4.9% (95% CI 4.8 to 5.0).

Key differences in characteristics of people in remission in 2019 compared to people who were not in remission in 2019 were older age (70% of people in remission were aged ≥65, compared to 54% of people not in remission); greater weight loss between diagnosis of diabetes and 2019; lower proportions with previous prescriptions of GLT (25% of people in remission had been prescribed GLT compared to 84% of people not in remission); lower mean HbA1c at diagnosis (note that although all people in our cohort had at least 1 HbA1c ≥48 mmol/mol from diagnosis to 2019, 38% of people in remission had an HbA1c <48 mmol/mol at diagnosis); and higher prevalence of previous history of bariatric surgery (although overall numbers

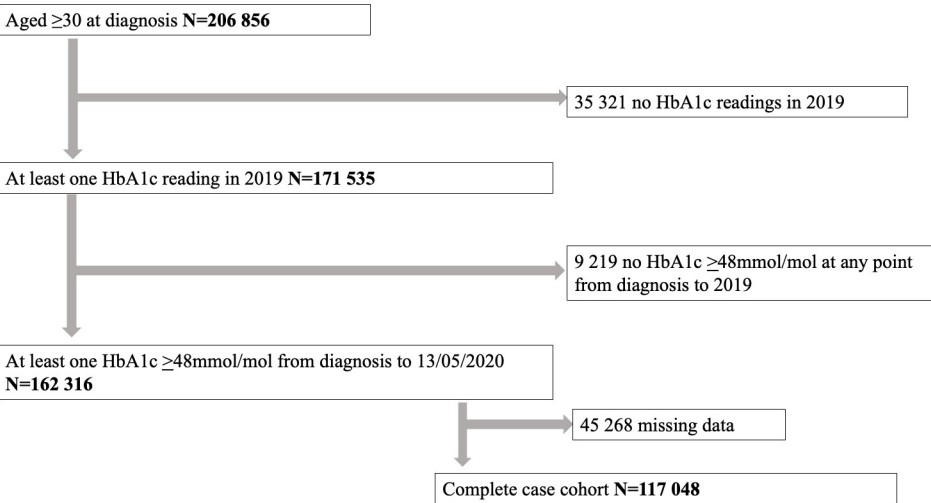

**Fig 1. Flow diagram showing the number of individuals with type 2 diabetes diagnosed in Scotland between January 1, 2004 and December 31, 2018 who were alive on December 31, 2019 and were registered with a Scottish health board who were eligible for inclusion in descriptive analyses/multiple imputation analysis and complete case analysis.**

of people with a history of bariatric surgery were small) (see Table 1). After stratification by age, the prevalence of remission in the study eligible population in 2019 was 3.2% in those aged <64 years and 6.0% in those aged ≥65 years. The highest remission prevalence was in the >75-year age group both before and after stratifying by sex (as described in S1 Fig) and HbA1c at diagnosis (as described in S2 Fig). The highest prevalence of remission was observed in the >75-year age group who had lost at least 15 kg since diagnosis of type 2 diabetes (14.5%) (see Fig 2). Median weight change from diagnosis to 2019 indicated weight loss in all sub-groups when stratified by sex, age, and remission status. People who were in remission in 2019 lost a median 6.5 kg of weight compared to 3.3 kg in people not in remission (S3 Fig). Women had greater median weight loss than men in most age groups with the exception of >75 year olds, for whom men who attained remission had greater median weight loss than women who attained remission (S3 Fig).

Characteristics associated with type 2 diabetes remission in 2019 in the primary logistic regression analysis of the imputed dataset were age in 2019, HbA1c at diagnosis, weight change, history of GLT, and history of bariatric surgery. Odds of remission were statistically significantly higher in people aged 65 to 74 and ≥75 years compared to people aged 46 to 54 years (Fig 3, S5 Table). Remission was inversely associated with median HbA1c at diagnosis. Within HbA1c categories, people with HbA1c <48 mmol/mol (<6.5%) at diagnosis had the highest odds of remission in 2019 compared to people with HbA1c 48 to 52 mmol/mol (6.5% to 6.9%) at diagnosis (Fig 3). The highest odds of remission were among people who had never been prescribed GLT compared to people who had been prescribed GLT. Remission was positively associated with weight loss. People who lost ≥15 kg from diagnosis to 2019 had the highest odds of being in remission in 2019 compared to people who gained 0 to 4.9 kg (Fig 3, S5 Table). People with a previous history of bariatric surgery had higher odds of remission compared with people with no previous history (OR 11.93 (95% 9.41 to 15.13) $P < 0.001$) (Fig 3). Bariatric surgery history and weight loss were each independently associated with remission. People in the least deprived quintiles were more likely to achieve remission than people in the most deprived quintile in univariable analysis, but these differences were not statistically significant in multivariable analyses (S5 Table). Multivariable analyses using complete case analyses showed similar results to the analyses using imputed data (S5 and S6 Tables, S4 and S5 Figs). When we excluded people with severe comorbidity (5.8% in the multiple imputed dataset) in a sensitivity analysis, we found that all the previously described associations between our covariates and remission in 2019 persisted with only small changes in ORs and CIs (Fig 4, S6 Table, S4 and S5 Figs).

People aged ≥75 years showed the most weight loss (Fig 2). Only 12.6% of our study population would have been eligible to participate in the DiRECT trial because exclusion criteria were people aged >65 years at diagnosis of diabetes (29.8% of our population), people with a duration of type 2 diabetes >6 years (63.7% of our population), people with BMI >45 kg/m² or <27 kg/m² (4.9% of our population), previous insulin use (7.9% of our population), and HbA1c ≥108 mmol/mol at diagnosis (1.5% of our population). The proportion of people in remission of type 2 diabetes in the subset of our population who were eligible for participation in the DiRECT trial was 4.7% (95% CI 4.4 to 5.0).

## Discussion

We found that the prevalence of type 2 diabetes remission in the study population was 4.8% in our primary analysis. Prevalence of remission was estimated to be 3.7% if none of the 17% of people with missing HbA1c data for 2019 was in remission. Conversely, including people without any record of HbA1c ≥48 mmol/mol (6.5%) in our cohort would increase the estimate of

**Table 1. Characteristics of people who were in remission in 2019 compared to people who were not in remission of type 2 diabetes among people in Scotland who had at least 1 HbA1c ≥48 mmol/mol (6.5%) after diagnosis of type 2 diabetes and had at least 1 HbA1c recorded in 2019.**

| | No remission (N = 154,606) | Remission (N = 7,710) | Total (N = 162,316) |
|---|---|---|---|
| Male | 88,664 (57.3%) | 4,124 (53.5%) | 92,788 (57.2%) |
| Age in 2019 (years) | | | |
| 30 to 44 | 5,977 (3.9) | 200 (2.6) | 6,177 (3.8) |
| 45 to 54 | 21,424 (13.9) | 666 (8.6) | 22,090 (13.6) |
| 55 to 64 | 42,208 (27.3) | 1,430 (18.5) | 43,638 (26.9) |
| 65 to 74 | 47,515 (30.7) | 2,342 (30.4) | 49,857 (30.7) |
| ≥75 | 37,482 (24.2) | 3,072 (39.8) | 40,554 (25.0) |
| Age at diagnosis (years) | | | |
| 30 to 44 | 20,557 (13.3) | 562 (7.3) | 21,119 (13.0) |
| 45 to 54 | 40,211 (26.0) | 1,190 (15.4) | 41,401 (25.5) |
| 55 to 64 | 49,137 (31.8) | 2,178 (28.2) | 51,315 (31.6) |
| 65 to 74 | 33,614 (21.7) | 2,482 (32.2) | 36,096 (22.2) |
| 75 plus | 11,087 (7.2) | 1,298 (16.8) | 12,385 (7.6) |
| Ethnicity | | | |
| White | 114,722 (74.2) | 5,721 (74.2) | 120,443 (74.2) |
| Other ethnic group | 7,377 (4.8) | 208 (2.7) | 7,585 (4.7) |
| Mixed or multiple ethnic group | 3,683 (2.4) | 189 (2.5) | 3,872 (2.4) |
| Missing or unavailable | 28,824 (18.6) | 1,592 (20.6) | 30,416 (18.7) |
| Duration of diabetes in 2019 (years) | | | |
| 0 to 1.9 | 10,335 (6.7%) | 224 (2.9%) | 10,559 (6.5%) |
| 2 to 5.9 | 45,113 (29.2%) | 3,300 (42.8%) | 48,413 (29.8%) |
| ≥6 | 99,158 (64.1%) | 4,186 (54.3%) | 103,344 (63.7%) |
| HbA1c at diagnosis (mmol/mol) | | | |
| <48 | 25,558 (18.5%) | 2,773 (38.2%) | 28,331 (19.4%) |
| 48 to 52 | 32,211 (23.3%) | 2,655 (36.5%) | 34,866 (23.9%) |
| 53 to 63 | 34,984 (25.3%) | 1,193 (16.4%) | 36,177 (24.8%) |
| 64 to 85 | 29,973 (21.6%) | 514 (7.1%) | 30,487 (20.9%) |
| ≥86 | 15,758 (11.4%) | 130 (1.8%) | 15,888 (10.9%) |
| Missing | 16,122 | 445 | 16,567 |
| BMI in 2019 (kg/m$^2$) | | | |
| Median (IQR) | 31.0 (27.5, 35.4) | 29.3 (25.9, 33.4) | 30.9 (27.4, 35.3) |
| Missing | 22,218 | 1,400 | 23,618 |
| BMI at diagnosis of diabetes (kg/m$^2$) | | | |
| Median (IQR) | 32.3 (28.7, 36.8) | 31.9 (28.4, 36.5) | 32.3 (28.7, 36.8) |
| Missing | 18,452 | 665 | 19,117 |
| Weight change (diagnosis to 2019) (kg) | | | |
| ≥5 gain | 11,719 (9.9%) | 227 (3.9%) | 11,946 (9.6%) |
| 0 to 4.9 gain | 23,530 (19.9%) | 717 (12.2%) | 24,247 (19.5%) |
| 0.1 to 4.9 loss | 34,265 (29.0%) | 1,439 (24.6%) | 35,704 (28.8%) |
| 5 to 9.9 loss | 25,401 (21.5%) | 1,458 (24.9%) | 26,859 (21.6%) |
| 10 to 14.9 loss | 13,006 (11.0%) | 894 (15.3%) | 13,900 (11.2%) |
| ≥15 loss | 10,397 (8.8%) | 1,123 (19.2%) | 11,520 (9.3%) |
| Missing | 36,288 | 1,852 | 38,140 |
| Smoking status at diagnosis | | | |
| Never smoked | 71,361 (46.2%) | 3,571 (46.3%) | 74,932 (46.2%) |
| Ever smoker | 83,058 (53.7%) | 4,133 (53.6%) | 87,191 (53.7%) |

(*Continued*)

**Table 1.** (Continued)

| | No remission (*N* = 154,606) | Remission (*N* = 7,710) | Total (*N* = 162,316) |
|---|---|---|---|
| Missing | 187 (0.1%) | 6 (0.1%) | 193 (0.1%) |
| SIMD 2016 (quintile) | | | |
| 1 (most deprived) | 36,352 (23.8%) | 1,594 (20.8%) | 37,946 (23.6%) |
| 2 | 35,550 (23.2%) | 1,668 (21.8%) | 37,218 (23.2%) |
| 3 | 31,473 (20.6%) | 1,631 (21.3%) | 33,104 (20.6%) |
| 4 | 27,744 (18.1%) | 1,449 (18.9%) | 29,193 (18.2%) |
| 5 (least deprived) | 21,883 (14.3%) | 1,312 (17.1%) | 23,195 (14.4%) |
| Missing | 1,604 | 56 | 1,660 |
| Previous metformin therapy | 126,175 (81.6%) | 1,808 (23.5%) | 127,983 (78.8%) |
| Previous sulphonylurea therapy | 59,619 (38.6%) | 447 (5.8%) | 60,066 (37.0%) |
| Previous glitazone therapy | 13,104 (8.5%) | 92 (1.2%) | 13,196 (8.1%) |
| Previous DPP-4 therapy | 37,344 (24.2%) | 105 (1.4%) | 37,449 (23.1%) |
| Previous SGLT-2 therapy | 22,409 (14.5%) | 28 (0.4%) | 22,437 (13.8%) |
| Previous GLP-1 therapy | 10,081 (6.5%) | 38 (0.5%) | 10,119 (6.2%) |
| Previous insulin therapy | 12,793 (8.3%) | 66 (0.9%) | 12,859 (7.9%) |
| Previous therapy with any GLT | 129,389 (83.7%) | 1,934 (25.1%) | 131,323 (80.9%) |
| History of severe comorbidity | | | |
| Dementia | 1,174 (0.8%) | 108 (1.4%) | 1,282 (0.8%) |
| Liver cirrhosis | 1,159 (0.7%) | 84 (1.1%) | 1,243 (0.8%) |
| CKD stage 5 | 985 (0.6%) | 69 (0.9%) | 1,054 (0.6%) |
| Cancer in last 5 years | 5,790 (3.7%) | 384 (5.0%) | 6,174 (3.8%) |
| Metastases in last 5 years | 1,015 (0.7%) | 78 (1.0%) | 1,093 (0.7%) |

BMI, body mass index; CKD, chronic kidney disease; DPP-4, dipeptidyl peptidase 4; GLP-1, glucagon-like peptide-1 receptor; GLT, glucose-lowering therapy; IQR, interquartile range; SGLT-2, sodium glucose cotransporter-2; SIMD, Scottish Index of Multiple Deprivation.

remission prevalence to 9.9%. The latter group includes people whose diagnosis of type 2 diabetes was based on glucose criteria alone, people who had recently moved to Scotland with a previous diabetes code, or who have been misclassified as having diabetes.

Prevalence of remission in the study cohort increased with age and was highest in women aged over 75 years (8.2%). Median weight decreased between diagnosis of type 2 diabetes and 2019 across the whole population, with people who were in remission in 2019 losing a median 6.5 kg of weight compared to 3.3 kg among people not in remission. We found that remission prevalence was lower in more socioeconomically deprived groups in univariable analyses. In multivariable models, factors associated with remission were age ≥65 years, HbA1c <48 mmol/mol (6.5%) at diagnosis, no history of GLT prescription, any weight loss from diagnosis to 2019, and a previous history of bariatric surgery. It is unsurprising that people who are closer to meeting the criteria for remission at diagnosis (HbA1c <48 mmol/mol and/or no prior GLT) have higher odds of remission. Older people develop diabetes at lower BMI and lower HbA1c than their younger counterparts [27]. It is possible that older people were diagnosed with diabetes with HbA1c values closer to diagnostic thresholds or after minor weight gain, and, therefore, only minor decreases in HbA1c or minor weight loss might be more likely to result in remission than among younger people. The lack of history of GLT prescription had a particularly strong association with remission. GLT is likely to be a marker for sustained hyperglycaemia or higher levels of glycaemia, but may also be an independent factor that decreases the risk of remission. Further research is required to investigate whether GLT has a

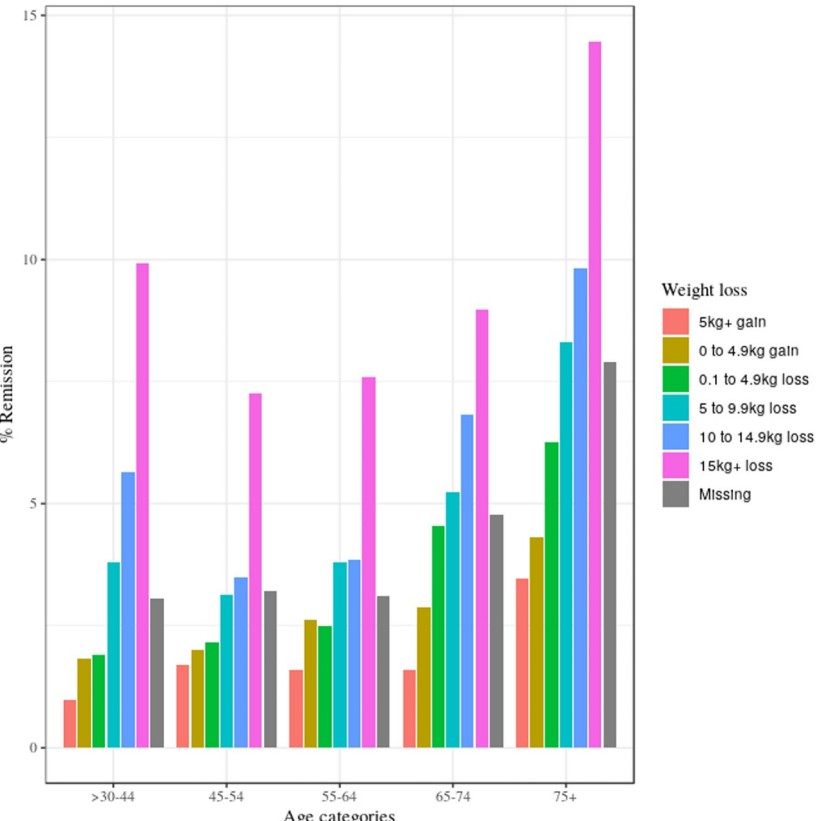

**Fig 2. Prevalence of remission of type 2 diabetes among people with type 2 diabetes in Scotland who had at least 1 HbA1c ≥48 mmol/mol (6.5%) after diagnosis of diabetes and had at least 1 HbA1c recorded in 2019 stratified by age group and change in weight from diagnosis to 2019.**

causal relationship with failure to achieve remission. Previous history of bariatric surgery also had a strong association with diabetes remission; however, bariatric surgery was rare in the study cohort (488 people), while there were almost 31,000 people with no history of GLT prescription. Bariatric surgery was associated with remission independently of weight loss, and this is consistent with previously documented findings that bariatric surgery, particularly malabsorptive bariatric procedures, is associated with remission of type 2 diabetes prior to significant weight loss [28]. All associations persisted after excluding people with severe comorbidity commonly associated with weight loss at diagnosis of diabetes, including people who may have been mistakenly assigned to the bariatric surgery group who were obese but whose indication for gastrectomy was for stomach cancer rather than obesity. A key strength of this study was that it was unlikely to have been unduly influenced by selection bias due to the use of a large contemporary whole population data registry. Another strength is that type of diabetes assigned by clinicians and the date of diagnosis held on SCI-Diabetes are validated for research purposes by an algorithm that uses prescription patterns, first mention of HbA1c ≥48 mmol/l (6.5%), retinopathy screening, and GLT prescription [2].

The study has a number of limitations. As for all studies using routine data, missing data were fairly common, although the proportions of missing data in all covariates were low enough to justify the use of multiple imputation and findings from both complete and imputed datasets were similar. We estimated prevalence under 3 sets of assumptions finding a range of 3.7% to 9.9%, although we believe the most appropriate estimate is of 4.8% from our primary

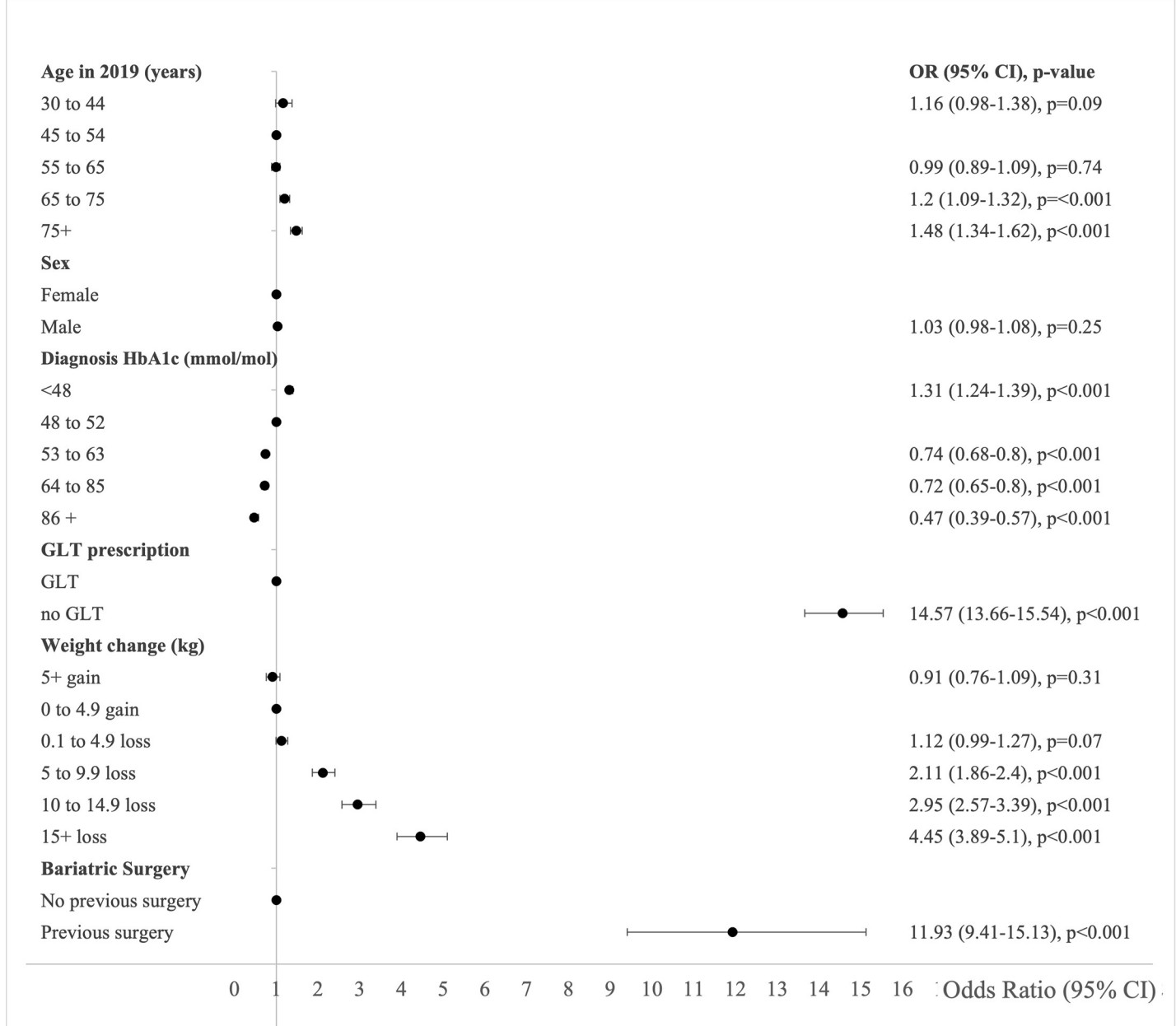

**Fig 3. ORs for remission of type 2 diabetes (95% CI) in Scotland 2019 derived from multivariable logistic regression model adjusted for all covariables listed on the plot with multiple imputation for missing data.** CI, confidence interval; GLT, glucose-lowering therapy; OR, odds ratio.

analysis. The prevalence estimates are influenced by our definition of type 2 diabetes remission. In our data, remission definition had to be based on HbA1c alone, because FPG is rarely measured after diagnosis of diabetes in the UK. Our remission estimate therefore does not account for people whose diabetes was diagnosed on the basis of fasting glucose values or 2-hr PG alone and who never had a record of HbA1c ≥48 mmol/mol (6.5%) after diagnosis of diabetes. A widely cited 2009 report by a multidisciplinary expert group was ambiguous as to whether both HbA1c and FPG measures are required to define remission [6], although a more recent position statement advises that remission can be defined in terms of HbA1c or FPG [4].

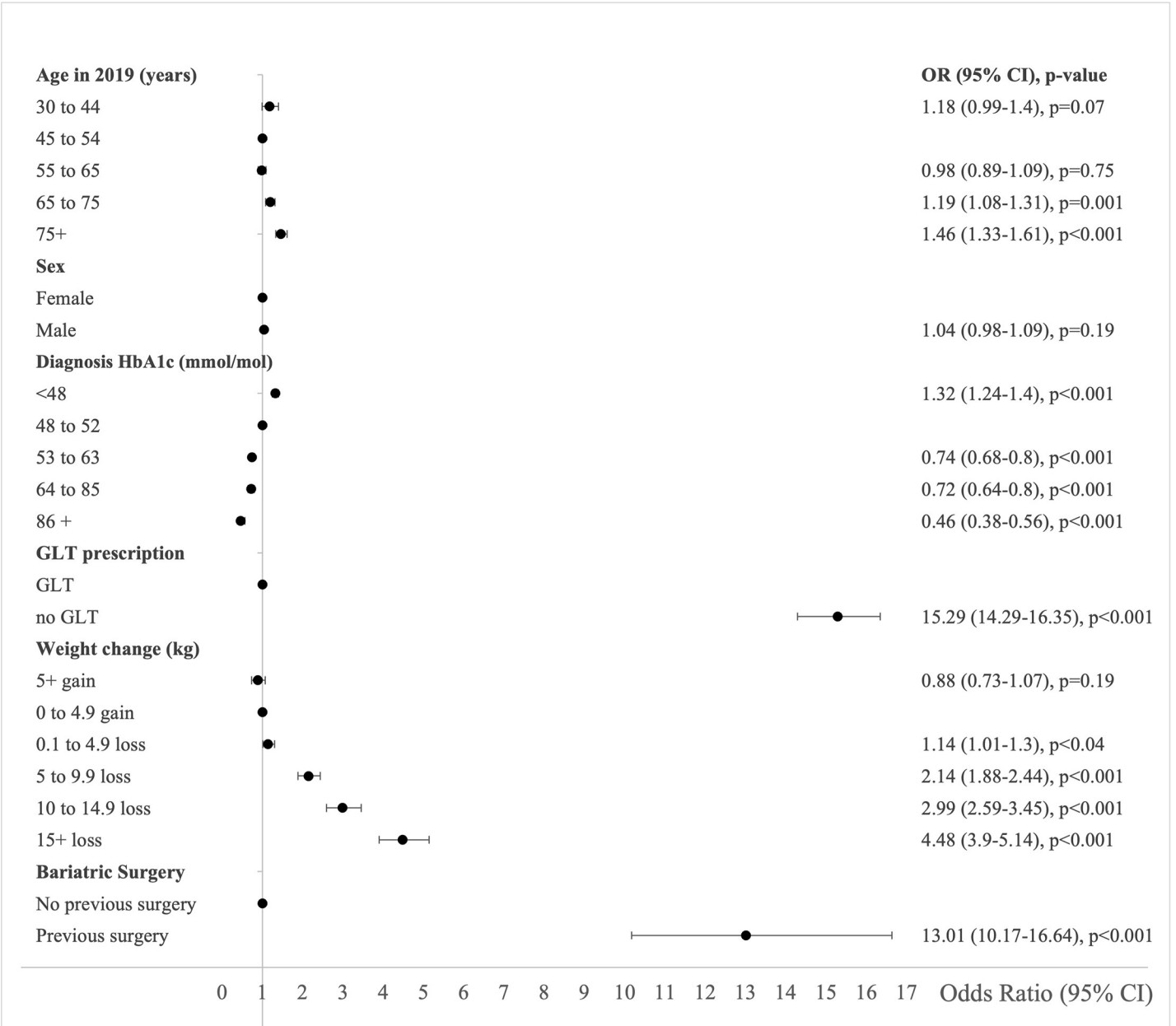

**Fig 4. ORs for remission of type 2 diabetes (95% CI) in Scotland 2019 derived from multivariable logistic regression model adjusted for all covariables listed on the plot with multiple imputation for missing data.** People with previous history of dementia, end-stage renal disease, liver cirrhosis, cancer, or metastases in the last 5 years removed. *N* = 152,934. CI, confidence interval; GLT, glucose-lowering therapy; OR, odds ratio.

This creates problems in defining remission for people with inconsistent values of glucose and HbA1c. Such ambiguities reflect the problems of defining a disease and its remission based on cutoffs in 2 continuous distributions (FPG and HbA1c). Moreover, HbA1c alone is not always appropriate to diagnose diabetes, and, similarly, may not always be appropriate to diagnose remission, for example, for people with certain haemoglobinopathies or anaemia of chronic disease [8,29]. Other potential options include use of estimated HbA1c; however, this is not established in routine clinical settings [7,30]. This highlights the need for a definitive and comprehensive consensus on the definition of type 2 diabetes remission that is clear for people

with type 2 diabetes and healthcare professionals in all settings [8]. We used a period of 1 year rather than 6 months to define remission as HbA1c tends to be measured on an annual basis in Scotland, particularly among people whose diabetes is well controlled. Using a 1-year duration of HbA1c below the diagnostic threshold for diabetes in the absence of GLT also facilitates comparison with previous studies on remission. A 1-year period was the most commonly used duration in definitions of remission (30% of definitions published in the research literature since 2009) [8]. However, the 2021 consensus report published during revision of this manuscript recommended that remission be defined at least 6 months after starting a lifestyle intervention by "HbA1c <6.5% (48 mmol/mol) measured at least 3 months after cessation of glucose-lowering pharmacotherapy" [30] (p. 1). This definition will be difficult to implement in routine clinical care in many settings when date of starting a lifestyle intervention may not be recorded and resources for measuring HbA1c more frequently than annually are limited. Finally, we may have underestimated the proportion of people who had received bariatric surgery as we did not have access to data on procedures performed in private hospitals, but bariatric surgery is relatively uncommon in the UK.

Karter and colleagues also used a diabetes registry to describe the epidemiology of remission in a Northern California population with health insurance, using a more conservative definition of remission (HbA1c <39 mmol/mol) based on the ADA diagnostic prediabetes criteria. They also excluded people who had bariatric surgery, and end of follow-up was December 31, 2011, prior to wider interest in the use of very low calorie diets to achieve remission of diabetes. They found a cumulative incidence of achieving remission over 7 years (1.60% [95% CI 1.53 to 1.68]) [20]. Similarities between our findings and Karter and colleagues' were that remission of type 2 diabetes was more common in people aged ≥65 years, people who were not prescribed GLT at diagnosis of diabetes, and people who had a lower HbA1c at diagnosis. Both studies found no association between BMI at diagnosis of type 2 diabetes and remission. Our study additionally found that weight loss was strongly associated with remission, which was not examined by Karter and colleagues. Duration was not significantly associated with remission in our study. Findings from trial settings have shown that remission is more likely in people with a short duration of diabetes. However, in routine clinical practice where diabetes reviews are approximately annual, it takes longer for people with short duration of diabetes to obtain all the necessary readings to meet criteria for prevalent remission. Karter and colleagues also found shorter duration of diabetes to be associated with incidence of remission in their cohort study because incidence studies are better suited to identifying the effect of shorter duration on remission than cross-sectional studies. The DiaRem score (developed to predict probability of remission within 5 years after bariatric surgery) also showed that age, HbA1c level, and GLT use were preoperative predictors of remission [31]. Future cohort studies could examine the covariates we found to be associated with remission to develop and validate a similar score that could predict remission after type 2 diabetes diagnosis. Our finding that remission is strongly independently associated with weight loss supports the results from the DiRECT trial where two-thirds of the 15% of trial participants that lost over 10 kg were in remission at 24 months [12]. In our study, people aged ≥75 years showed the highest prevalence of remission (and weight loss) (Fig 2). This fits with previous research in adults over 70 years with prediabetes; regression to normoglycaemia; or death occurred more frequently than progression of prediabetes to diabetes [32]. Possible contributing factors include unintentional weight loss as a consequence of frailty or progressive multimorbidity and lower HbA1c at diagnosis of diabetes than younger people [25], thus closer to the diagnostic threshold with higher sensitivity to smaller fluctuations.

Our prevalence estimates suggest that a reasonably large proportion of people achieve remission of type 2 diabetes in routine clinical care outside trial or bariatric surgery settings.

The immediate implications for practice are that these people should be recognised and coded appropriately so they can be given adequate support and followed up to ensure continued care consistent with diabetes management guidelines. It is important to recognise that remission of diabetes may not be permanent. The Look Action for Health in Diabetes (AHEAD) trial among overweight and obese United States adults with type 2 diabetes found that, of the participants who attained remission of type 2 diabetes, a third of the group that received an intensive lifestyle intervention, and 40% to 50% of the control group returned to type 2 diabetes status each year [10]. Further research is needed to investigate the association between remission and complications of diabetes including mortality. Particularly, the effect of different durations of sustained remission and complications. The Look AHEAD trial was stopped after 8 to 11 years of follow-up as the intensive lifestyle intervention did not decrease risk of CV events; however, the effect of remission of diabetes in this population was not described [10]. There may be a group of people in remission in whom screening intervals for complications of diabetes can safely be lengthened. This could reduce the burden of chronic disease for both patients and health services.

Our findings highlight that there is no differentiation between intentional and nonintentional weight loss in current definitions of remission, and people who attain remission by unintentional weight loss caused by severe illness could still be defined as being in remission of type 2 diabetes [4,8]. Therefore, unless forthcoming revisions of the international consensus definition of remission specifies that remission or weight loss must be intentional, then greater clarity is needed as to how older people with type 2 diabetes who meet remission criteria by unintentional weight loss are coded and managed clinically and how their data should be handled in observational studies. There may be a tendency to remission (as currently defined) with age, given the association between age and increasing number and severity of comorbidities that may predispose to unintentional weight loss. This may influence associations between attainment of remission and mortality. Our findings suggest that people who were older than the age eligibility criteria for the DiRECT trial have the capacity to achieve remission of type 2 diabetes, although we were not able to establish whether weight loss was intentional or unintentional. Having established that remission is possible using nonsurgical interventions, future trials could now broaden their inclusion criteria to explore the effectiveness of interventions in a more representative sample of the type 2 diabetes population. Remission definitions have evolved alongside interventional bariatric and dietary trials where unintentional weight loss (especially among older people) may have been less common due to exclusion criteria. However, as remission is implemented in routine care, further consideration is needed as to whether older people and/or people with unintentional weight loss constitute a subset of people in remission or whether future iterations of the remission definition should exclude these groups by specifying that weight loss must be intentional. Such a definition would be difficult to implement in routine healthcare records.

Our results may also be relevant in supporting people to attain remission in the future; recognising characteristics that make type 2 diabetes remission more likely could facilitate discussions about remission, weight loss, or bariatric surgery. Further cohort studies could build on our results to develop and validate risk scores for remission of diabetes. Our results suggest that supporting clinicians to discuss remission and to refer to weight management services with early or diet-controlled diabetes is a rational approach. However, more data are needed to establish whether a very low calorie diet approach will be effective or appropriate in people that have been excluded from trials. Our estimates for prevalence of type 2 diabetes remission reflect the period prior to the widespread introduction of the use of very low calorie diets in routine clinical care and the COVID-19 pandemic so further estimates of remission prevalence estimates will be required in the future.

In conclusion, we have shown that approximately 5% of people with type 2 diabetes in Scotland in 2019 appear to be in remission and that history of never having GLT, weight loss since diagnosis, HbA1c <48 mmol/mol (6.0%) at diagnosis, age ≥65 years, and previous bariatric surgery are associated with remission. Our findings provide a useful basis for the evaluation of the multifactorial approaches to both remission and prevention of diabetes that are currently being introduced and also highlight the need for guidelines to support definition of remission, management, and follow-up of people that achieve remission.

## Supporting information

**S1 Table. STROBE Checklist.** STROBE, STROBE, Strengthening The Reporting of OBservational Studies in Epidemiology.
(DOCX)

**S2 Table. OPCS-4 procedure codes used in combination with ICD-10 diagnosis codes for obesity and type 2 diabetes by Public Health Scotland to identify people with a history of bariatric surgery.** Number of procedures for people with an obesity code are given against each OPCS code for the present analysis. ICD-10, International Classification of Diseases-10th revision; OPCS-4, Office of Population Censuses and Surveys Classification of Interventions and Procedures version 4.
(DOCX)

**S3 Table. ICD-10 and ICD-9 codes used to identify comorbidities.** ICD-9, International Classification of Diseases-Ninth revision; ICD-10, International Classification of Diseases-10th revision.
(DOCX)

**S4 Table. Proportions of missing data within the variable of interest.** Odds of missing data in people with type 2 diabetes in Scotland diagnosed ≥30 years of age who had at least 1 HbA1c ≥48 mmol/mol (6.5%) after diagnosis of diabetes and who were alive and had at least 1 HbA1c recorded in 2019.
(DOCX)

**S5 Table. Unadjusted and adjusted odds for remission of type 2 diabetes from logistic regression model using complete case and multiple imputation datasets [1] *N* = 162,316 (multiple imputation) and *N* = 117,048 (CCA).** Adjusted for all variables in the model (listed in first column of the table). CCA, Complete case analysis.
(DOCX)

**S6 Table. Sensitivity analysis with 10,846 people with comorbidities removed.** Unadjusted and adjusted odds for remission of type 2 diabetes from logistic regression model using complete case and multiple imputation datasets. [1] Adjusted for all variables in the model (listed in first column of the table). *N* = 152 934 (multiple imputation) and *N* = 110,814 (CCA). CCA, complete case analysis.
(DOCX)

**S1 Fig. Prevalence of remission of type 2 diabetes among the Scottish type 2 diabetes population of people who had at least 1 HbA1c ≥48 mmol/mol (6.5%) after diagnosis of diabetes and had at least 1 HbA1c recorded in 2019.** Stratified by sex and age in 2019.
(DOCX)

**S2 Fig. Prevalence of remission of type 2 diabetes among the Scottish type 2 diabetes population of people who had at least 1 HbA1c ≥48 mmol/mol (6.5%) after diagnosis of**

**diabetes and had at least 1 HbA1c recorded in 2019 by age in 2019.**
(DOCX)

**S3 Fig. Median weight change (kg) from diagnosis of type 2 diabetes to 2019 according to remission ($n$ = 7,710) and nonremission status ($n$ = 154,606).**
(DOCX)

**S4 Fig. ORs for remission of type 2 diabetes (95% CI) in Scotland 2019 derived from complete case analysis logistic regression model adjusted for all covariables listed on the plot.**
$N$ = 117,048. CI, confidence interval; OR, odds ratio.
(DOCX)

**S5 Fig. ORs for remission of type 2 diabetes (95% CI) in Scotland 2019 derived from complete case analysis logistic regression model adjusted for all covariables listed on the plot.**
People with previous history of dementia, end-stage renal disease, liver cirrhosis, cancer, or metastases in the last 5 years removed. $N$ = 110,814. CI, confidence interval; OR, odds ratio.
(DOCX)

## Author Contributions

**Conceptualization:** Mireille Captieux, Sarah H. Wild.

**Data curation:** Mireille Captieux, Kelly Fleetwood, Sarah H. Wild.

**Formal analysis:** Mireille Captieux, Kelly Fleetwood, Bruce Guthrie, Sarah H. Wild.

**Funding acquisition:** Mireille Captieux.

**Investigation:** Mireille Captieux, Bruce Guthrie, Sarah H. Wild.

**Methodology:** Mireille Captieux, Kelly Fleetwood, Bruce Guthrie, Sarah H. Wild.

**Project administration:** Mireille Captieux, Sarah H. Wild.

**Supervision:** Bruce Guthrie, Sarah H. Wild.

**Writing – original draft:** Mireille Captieux.

**Writing – review & editing:** Mireille Captieux, Kelly Fleetwood, Brian Kennon, Naveed Sattar, Robert Lindsay, Bruce Guthrie, Sarah H. Wild.

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
