## [Editor Report · Decision Letter 0]

9 Jun 2021

Dear Dr Captieux, 

Thank you for submitting your manuscript entitled "Epidemiology of type 2 diabetes remission in Scotland in 2019: a cross-sectional population-based study" for consideration by PLOS Medicine.

Your manuscript has now been evaluated by the PLOS Medicine editorial staff and I am writing to let you know that we would like to send your submission out for external assessment.

However, before we can send your manuscript for assessment, we need you to complete your submission by providing the metadata that is required for full assessment. To this end, please login to Editorial Manager where you will find the paper in the 'Submissions Needing Revisions' folder on your homepage. Please click 'Revise Submission' from the Action Links and complete all additional questions in the submission questionnaire.

Please re-submit your manuscript within two working days, i.e. by Jun 11 2021 11:59PM.

Once your full submission is complete, your paper will undergo a series of checks in preparation for further assessment. 

Kind regards,

Richard Turner, PhD

rturner@plos.org

---

## [Decision Letter · Decision Letter 1]

4 Aug 2021

Dear Dr. Captieux,

Thank you very much for submitting your manuscript "Epidemiology of type 2 diabetes remission in Scotland in 2019: a cross-sectional population-based study" (PMEDICINE-D-21-02522R1) for consideration at PLOS Medicine. 

Your paper was discussed with an academic editor with relevant expertise and sent to independent reviewers, including a statistical reviewer. The reviews are appended at the bottom of this email and any accompanying reviewer attachments can be seen via the link below:

[LINK]

In light of these reviews, we will not be able to accept the manuscript for publication in the journal in its current form, but we would like to invite you to submit a revised version that addresses the reviewers' and editors' comments fully. You will appreciate that we cannot make a decision about publication until we have seen the revised manuscript and your response, and we expect to seek re-review by one or more of the reviewers. 

We hope to receive your revised manuscript by Aug 25 2021 11:59PM. Please email us (plosmedicine@plos.org) if you have any questions or concerns.

Please let me know if you have any questions, and we look forward to receiving your revised manuscript. 

Sincerely,

Richard Turner PhD

Senior editor, PLOS Medicine

rturner@plos.org

At line 37, please add "We carried out a ..." or similar.

In the abstract, please quote aggregate demographic details for study participants. 

Please add a new final sentence to the "Methods and findings" subsection of your abstract, beginning "Study limitations include ..." or similar and quoting 2-3 of the study's main limitations. 

At line 55, please begin the sentence with "In this study, we found that ..." or similar. 

Again at line 55, please avoid statements of the type "almost X" in favour of quoting the actual number or proportion. 

Please remove the information on funding from the abstract page. In the event of publication, this information will be included in the article metadata, via entries in the submission form. 

After the abstract, we will need to ask you to add a new and accessible "Author summary" section in non-identical prose. You may find it helpful to consult one or two recent research papers published in PLOS Medicine to get a sense of the preferred style. 

Early in the Methods section, please state whether the study had a protocol or prespecified analysis plan, and if so attach the document as a supplementary file, referred to in the text. Please highlight analyses that were not prespecified. 

Where available, please quote p values alongside 95% CI. 

Please remove the footnotes (these points can be integrated into the text, we suspect). 

Throughout the text, please remove the spaces from the square brackets containing reference call-outs.

Please include a completed checklist for the most appropriate reporting guideline, e.g., STROBE or RECORD, as an attachment with your revision, labelled "S1_RECORD_Checklist" or similar and referred to as such in your Methods section. 

In the checklist, please refer to individual items by section (e.g., "Methods") and paragraph number rather than by line or page numbers, as the latter generally change upon publication. 

Comments from Academic editor:

I read this manuscript with interest. As pointed out by the authors and reviewers, diabetes remission is an area of much interest and public health significance. Hence this study, one of the very few that has examined prevalence of diabetes remission using population-based data, is of great interest. Although the findings are not too surprising, the results, highlighting the potential possibility of diabetes remission outside of intensive interventions in trial settings using VLC diets, is still of much public health interest.

I do, however, have some concerns about the analysis. In addition to points raised by the reviewers, I agree with one of the reviewers that on close inspection, many of the procedures included in Supplementary table 1 are not procedure codes appropriate for bariatric surgery, but include many procedural codes associated with major GI surgery (e.g. gastrectomy for stomach cancer). This raises the issue of the marked weight loss and DM remission being related to co-existing malignancy and the associated weight loss due to the underlying disease or GI surgery.

The authors would need to provide a detailed explanation to address this concern, which would very much impact on the validity of the analysis. Another major concern, which has not been discussed or addressed, is the potential remission of diabetes that occurs when patients develop significant chronic kidney disease or end stage renal disease, whereby medications are no longer necessary, and A1c may have become normal (or low), due to both diabetes remission as well as co-existing anaemia of chronic disease. A sensitivity analysis to ensure that patients with concomitant malignancy or chronic kidney disease have been excluded would be necessary, in my view.

Comments from the reviewers:

*** Reviewer #1: 

This study utilises Scottish population data to explore the prevalence of remission of type 2 diabetes. The authors are able to assess a number of key measures including age, diabetes duration, BMI and HbA1c at diagnosis, weight loss, diabetes treatment and presence of other diagnoses to identify the measures associated with remission (defined as >1 year with an HbA1c <48 mmol/mol). This is an important study as it tries to quantify the prevalence of remission in the general diabetes population. Thus, moving the conversation forward as to how we address remission in both epidemiology and clinical practice.

I have some issues with the presentation of the results and areas which I think need to be more clearly described.

1. I found table S3 hard to follow and would appreciate greater explanations as to how to interpret the displayed data. 

2. Figure S2 shows the prevalence of remission by age and HbA1c group at diagnosis but the figure header says the results are by duration of type 2 diabetes and age in 2019. I can not see how duration is included in this figure.

3. Greater weight loss is associated with remission - did you consider looking at weight change as a percentage of baseline weight? Similarly did change in BMI show any association?

4. The cut-point of over a year of HbA1c values <48 mmol/mol is valid due to the annual nature of HbA1c measures among people with type 2 diabetes. Did you look at people who failed to meet the criteria because their last HbA1c was just under a year (ie 11 months rather than 12 months)? Would a cut point of at least 10 months change your results?

5. Any data on complications - specifically retinopathy around the time of diagnosis which indicates potentially delayed diagnosis?

6. You do not find that diabetes duration is associated with risk of remission after adjustment for other covariates. I think this is worthy of some discussion as it seems intuitive that remission is easier in people with new onset diabetes but that is not the case here.

7. The association with use of GLT is interesting. I would appreciate some discussion of your interpretation of this - do you think this association is because GLT correlates with sustained hyperglycaemia and symptoms or could it also suggest that treatment with hypoglycaemic agents is deleterious to chances of remission? I appreciate your desire not to overly interpret your findings but I think some discussion is warranted.

8. Can you include some comments on the ethnic make up of your study? It being Scotland I presume it is primarily Caucasian but do you have any estimate on the prevalence of Asian and Black participants? 

9. Have you any data for the length of remission observed in the population. How often do you see a period of an HbA1c <48mmol/mol for at least a year followed by HbA1c values >48mmol/mol. Is the remission you observe likely to be prolonged or more akin to the honeymoon period sometimes observed in early diabetes?

*** Reviewer #2 (statistical reviewer): 

This cross-sectional study aims to estimate the prevalence of remission of type 2 diabetes in all adults in Scotland aged >30 years diagnosed with type 2 diabetes and alive on 31/12/2019.

Comments:

"Factors associated with remission were: older age (OR 1.48 (95% CI 1.34-1.62) for people aged >75 years compared to 45-54 year group), HbA1c <48mmol/mol at diagnosis (OR 1.31 (95%CI 1.24-1.39) compared to 48-52mmol/mol), no previous history of glucose lowering therapy (OR 14.6, 95%CI 13.7-15.5), weight loss from diagnosis to 2019 (OR 4.45 (95%CI 3.89-5.10) for >15kg of weight loss compared to 0-4.9kg weight gain) and previous bariatric surgery (OR 11.9, 95%CI 9.41-15.1)."

Did the authors consider investigating interactions between factors?

"For the purposes of this analysis we defined remission on the basis of all HbA1c values being <48mmol/mol in the absence of glucose lowering treatment (GLT) for a duration of >365 days before the date of the last recorded HbA1c in 2019, that is on the basis of at least two HbA1c values <48mmol/mol at an interval of at least 365 days. "

Did the authors consider undertaking any sensitivity analyses on different definitions of remission?

"Demographic variables recorded at date of diagnosis of type 2 diabetes were: sex and age (categorised into the following groups: 30 to 44, 45 to 54, 55 to 64, 65 to 74 and 75 and over 150 years)."

Can the authors please confirm if age was categorised at the point of data collection, or during data processing? If the latter, did the authors consider including age as a continuous variable in their analyses?

"People with no record of an HbA1c >48mmol/mol (6.5%) after diagnosis of diabetes or no HbA1c recorded during 2019 were excluded initially but included for sensitivity analyses for estimates of prevalence of remission."

The authors have conducted a valuable sensitivity analysis which helps to demonstrate the robustness in their study findings.

The authors have applied comprehensive and rigorous statistical modelling methods, which they describe clearly and concisely. They have appropriately handled missing data by applying MICE, and have suitably conducted a complete case sensitivity analysis. The main study limitations have been thoroughly explored in the discussion section.

*** Reviewer #3: 

This paper has addressed the rates of remission of T2D in the Scottish adult population. Their main findings were that age ≥65, no previous glucose lowering medication, weight loss since the diagnosis of T2D and previous bariatric surgery were associated with a higher chance of being one of the 5% of patients with T2D who achieved remission. 

There are some issues that need clarification. 

1. The finding that previous bariatric surgery was associated with remission of T2D is somewhat confusing. If I understand the methods correctly these patients underwent bariatric surgery before the diagnosis of T2D. What was the time interval between the surgery and diagnosis of T2D? It would seem likely that it was short as it has been shown that new cases of T2D after bariatric surgery is unusual (particularly with modern procedures). Thus, the remission of T2D was not associated with previous bariatric surgery but the effect of the surgery, which is consistent with the effect of bariatric surgery on T2D. Why was not bariatric surgery after the diagnosis of T2D studied?

2. In addition, the surgical codes presented in Table S1 are confusing. I am not familiar with the OPCS codes but bases on the written description many of the codes used do not represent bariatric surgery, but other procedures performed on the upper gut. Many are procedures done for cancer surgery. Some will have a similar impact on changing the anatomy of the foregut as bariatric surgery with similar impact on gut physiology as bariatric surgery. Yet, they are not bariatric surgery and surgery for cancer comes with other confounding factors. 

3. The authors emphasize that weight loss (achieved without surgery) is an important factor. However, it needs to be pointed out that persistent weight loss without bariatric surgery is difficult to achieve. The findings are also necessary to put in the perspective of impact of remission on hard endpoints. Lessons learned from the look AHEAD study.

*** Reviewer #4: 

Epidemiology of type 2 diabetes remission in Scotland in 2019: a cross-sectional population-based study

Summary/ Overall Comments

Thank you for the opportunity to review this submission. The authors conducted a very well-designed study describing the prevalence of T2DM remission in a large cohort of patients and uses sound statistical methods to describe the predictors of remission. The authors capitalize on a very extensive national database with great details and follow-up data on patients with diabetes in Scotland. This study reflects a very effective use of this valuable database to answer important clinical questions to guide patient care and future research. The manuscript is well-written and the rationale, methods, and findings are easy to follow throughout the manuscript. I commend the authors on this excellent work. I only have a few questions and minor comments regarding the study.

Abstract

- Well-written and clearly summarizes the study.

Introduction

- Well-written Introduction section, provides the right amount of background and literature review to help readers understand the purpose of the study.

- The authors describe the study aims on Lines 105-107. I would suggest that one of the aims of the study was to identify predictors of T2DM remission in their multivariable regression analysis, in addition to their stated aim of comparing the characteristics of patients with vs. without remission.

Methods

- The overall study methods, eligibility criteria, definition of variables, and statistical methods are well-described.

- Thank you for including a section describing patient/ public involvement. Although patients and the public were not involved in the design of the study, inclusion of this section and details is of utmost importance and unfortunately, often neglected in manuscripts.

- Approach to missingness was described very well.

- Please describe how the decision regarding which covariates to include was derived? If based on existing literature on the predictors of T2DM remission, please cite all relevant papers. Were the covariates selected based on a series of studies that found an association with each of the selected covariates separately or studies that evaluated all the covariates? Or were these covariates decided solely based on clinical rationales? Please describe.

- Many of the covariates selected are part of the DiaREM score which consists of similar, albeit fewer variables, that have been developed and validated to predict T2DM remission following RYGB surgery (PMID: 24579062, PMID: 28349641, PMID: 26537267). Although this predictive score is limited to remission post bariatric surgery, perhaps the authors can cite this score and/ or any other scores in a more general setting unrelated to bariatric surgery, that consist of a similar set of predictors.

- I'm curious as to why the authors chose to treat age as a categorical variable in its inclusion as a covariate, rather than continuous despite this being available to the authors. Would it not be more optimal to maintain continuous variables as continuous whenever possible?

- Why was weight (mentioned in Line 157) mentioned as a covariate? Would BMI, already a covariate, be a more accurate reflection in this case as it accounts for height?

- Did the database contain information about the type of bariatric surgery that patients underwent? Were there any gastric restrictive surgeries (e.g., sleeve gastrectomy or bands) or were they all malabsorptive (e.g., RYGB)? If any details on these are available in the database or perhaps based on local surgical practices during the timeframe of the study (perhaps only certain types of bariatric procedures are offered in Scotland?), it would be great to describe these.

Results

- The Results section is well-structured, beginning with a detailed description of the sample and flows nicely through description of the various findings.

- Great use of figures to summarize a large number of findings.

- Figure 1 - This figure should be mentioned in the first part of the Results section when the included population is first described, rather than in the Methods section.

- Figure 3 - Very helpful forest plot. I would suggest adding labels to the x-axis for easier interpretation for readers.

- Was bariatric surgery associated with higher SES? Perhaps this association is strong enough to account for the fact that SES was not a predictor in the multivariable analysis (once SES was factored in)?

- Please briefly describe the DiRECT trial that is discussed in the final paragraph of the Results section.

- It seems that bariatric surgery was a predictor of T2DM remission independent of weight loss since both were statistically significant predictors in the multivariable model. If that's the case, this is an interesting finding that may be worth explicitly mentioning in the Results & discussing further in the Discussion section.

Discussion

- This section provides adequate details to help contextualize the study findings yet is succinct and clear to follow.

- Minor comment - Lines 310-311 - This statement "... inversely associated with socioeconomic deprivation..." suggests that remission was associated with higher SES but in its current form describing the inverse relationship with socioeconomic deprivation, the statement is somewhat difficult to understand and could be simplified.

***

[LINK]

---

## [Decision Letter · Decision Letter 2]

28 Sep 2021

Dear Dr. Captieux,

Thank you very much for re-submitting your manuscript "Epidemiology of type 2 diabetes remission in Scotland in 2019: a cross-sectional population-based study" (PMEDICINE-D-21-02522R2) for consideration at PLOS Medicine.

I have discussed the paper with editorial colleagues and our academic editor, and it was also seen again by four reviewers. I am pleased to tell you that, provided the remaining editorial and production issues are fully dealt with, we expect to be able to accept the paper for publication in the journal.

[LINK]

Please let me know if you have any questions, and we look forward to receiving the revised manuscript.   

Sincerely,

Richard Turner, PhD

rturner@plos.org

Requests from Editors:

Please finalize the arrangements for data deposition, and add the details to the data statement (submission form). 

We suggest removing "who wish" at line 32 (abstract). 

Please convert the three subtitles into boldface text; and add bullets to the individual points in the Author Summary.

Please remove the point at line 70 ("Remission of type 2 diabetes ...").

Please trim the point at line 101 (generally we anticipate three points per subsection, each extending to one to two short sentences).

Throughout the text, please incorporate single spaces before the reference call-outs (e.g., "... diets [10,11].").

In the abstract and at any other instances, please use square brackets within parentheses where needed, e.g., "(OR 1.48 [95% CI 1.34-1,62]...)".

Please remove reference 23 (this could be quoted as a "personal communication" in the text, but this may not be necessary). 

Noting reference 25, can a URL or publication details be added?

Please break the STROBE checklist out into a separate attached file, labelled "S1_STROBE_Checklist" or similar and referred to as such in the text. 

Comments from Reviewers:

*** Reviewer #1: 

I am happy that the authors have addressed my prior comments.

*** Reviewer #2: 

The authors have satisfactorily considered and responded to each comment in turn.

*** Reviewer #3: 

I am happy with the changes made

*** Reviewer #4: 

I thank the authors for considering my comments on this manuscript. They have addressed my comments and I have no further suggestions for the manuscript. Thank you for the opportunity to review this interesting study.

***

[LINK]

---

## [Editor Report · Decision Letter 3]

30 Sep 2021

Dear Dr Captieux, 

On behalf of my colleagues and the Academic Editor, Dr Ma, I am pleased to inform you that we have agreed to publish your manuscript "Epidemiology of type 2 diabetes remission in Scotland in 2019: a cross-sectional population-based study" (PMEDICINE-D-21-02522R3) in PLOS Medicine.

Prior to final acceptance, as a condition of publication please ensure that anonymized study data are made available and the access details provided in the article metadata so as to comply fully with PLOS' data policy (https://journals.plos.org/plosmedicine/s/data-availability).

Please also read through the text and correct any typos, noting "... a limited subset" at line 55, and "... for which there was" at line 183.

PRESS

Sincerely, 

Richard Turner, PhD 

rturner@plos.org